# The Tannins from *Sanguisorba officinalis* L. (Rosaceae): A Systematic Study on the Metabolites of Rats Based on HPLC–LTQ–Orbitrap MS^2^ Analysis

**DOI:** 10.3390/molecules26134053

**Published:** 2021-07-02

**Authors:** Jingjing Tu, Qiaoling Li, Benhong Zhou

**Affiliations:** 1Department of Pharmacy, Renmin Hospital of Wuhan University, Wuhan 430060, China; tujj@nicemice.cn (J.T.); Tu_jingjing12@163.com (Q.L.); 2School of Pharmaceutical Sciences, Wuhan University, Wuhan 430071, China

**Keywords:** *Sanguisorba* tannins, HPLC–LTQ–Orbitrap MS^2^, in vivo, in vitro, metabolites

## Abstract

*Sanguisorba* tannins are the major active ingredients in *Sanguisorba ofJicinalis* L. (Rosaceae), one of the most popular herbal medicines in China, is widely prescribed for hemostasis. In this study, three kinds of tannins extract from *Sanguisorba officinalis* L. (Rosaceae), and the metabolites in vivo and in vitro were detected and identified by high-pressure liquid chromatography, coupled with linear ion trap orbitrap tandem mass spectrometry (HPLC–LTQ–Orbitrap). For in vivo assessment, the rats were administered at a single dose of 150 mg/kg, after which 12 metabolites were found in urine, 6 metabolites were found in feces, and 8 metabolites were found in bile, while metabolites were barely found in plasma and tissues. For in vitro assessment, 100 μM *Sanguisorba* tannins were incubated with rat liver microsomes, liver cytosol, and feces, after which nine metabolites were found in intestinal microbiota and five metabolites were found in liver microsomes and liver cytosol. Moreover, the metabolic pathways of *Sanguisorba* tannins were proposed, which shed light on their mechanism.

## 1. Introduction

The *Sanguisorba officinalis* L. (Rosaceae), with a long history of cultivation, is widely distributed in Asia, Western Europe, and North America and is specifically native to the temperate zones of the Northern Hemisphere, a kind of plant that is available for medicine in China, Europe, Japan, and Korea [1]. It is a member of subfamily *Rosoideae* and family *Rosaceae*, and all parts of the plant can be used as an effective astringent, but the root is the most active [2]. The dried root of *Sanguisorba* is recorded in various versions of Chinese Pharmacopoeia, European Pharmacopoeia, Russian Pharmacopoeia [3,4]. Since the *Sanguisorba* contains phenolic compounds, such as tannic acid, with a function of “cooling blood for hemostasis, detoxification, and astringency”, it has been used for metrorrhagia and metrostaxis, hematemesis, hematochezia, bleeding hemorrhoids, pudendum eczema, menopathy, and leukorrheal diseases [5]. The main constituents include triterpenoid saponins, flavonoids, phenols, etc., among which the phenols are highest and are quite active in the treatment for increasing white blood cell, along with having anticancer, antioxidant, and antibacterial benefits [6].

According to the chemical structure of tannins, they can be divided into three categories: (i) hydrolyzable tannins, which generally refer to tannins that can produce gallic acid or various gallic acid polymers (CG, S-HHDP, DHDDP, S, S-gallagyl, lactone valoneoyl) and glucose after hydrolysis; (ii) condensed tannins, which are a class of compounds in which catechins or their derivatives and other flavan-3-ols are polymerized with C–C bonds [7,8]. Since there is no glycoside bond and ester bond in their structure, they cannot be hydrolyzed by acid or base [9]; and (iii) complex tannins, which are composed of the components of condensed tannins, flavan-3-ols, and the hydrolyzed tannins by carbon–carbon bonding. These tannins were first isolated from the *Fagaceae* [10] and have been widely found in plants containing both hydrolyzable tannins and condensed tannins [11].

At present, the limited studies available mainly focus on the metabolites and tissue distribution of Ziyuglycoside I and II [12], the major active ingredients isolated from *Sanguisorba officinalis*, with few studies about *Sanguisorba* tannins.

In the present study, high-performance liquid chromatography, coupled with high-resolution mass spectrometry (HPLC–HRMS), has been used as an effective and reliable analytical tool to identify metabolites due to its high resolution and high sensitivity [13,14]. Metabolism research can identify metabolites and major metabolic pathways, which can help to understand the pharmacological mechanisms of food and drugs [15]. Recently, the intestinal microbiota has been considered to be a “hidden organ” of the body [16], which can be regarded as a systematic research approach to study the intestinal microbiota metabolism in vitro [17]. In addition, *Sanguisorba* tannins can be metabolized by liver microsomes, cytosol, and rat primary hepatocytes in vitro [18], which has been regarded as a valuable model to study food and drug metabolism in vitro due to the biotransformation organ of the liver [19].

Therefore, in this study, we conducted a comprehensive in vivo–in vitro investigation on the metabolism of *Sanguisorba* tannins using HPLC–LTQ–Orbitrap MS^2^. In vitro, *Sanguisorba* tannins mediated metabolism was studied in the intestinal microbiota and liver microsomal culture system of rats. In vivo, *Sanguisorba* tannins metabolites were characterized in various biological samples, including urine, feces, plasma, bile, and various tissues, after oral administration of *Sanguisorba* tannins in adult rats.

## 2. Results

### 2.1. Establishment of the Analytical Strategy

In this study, an effective strategy was taken for metabolite identification of *Sanguisorba* tannins in vivo and in vitro using HPLC–LTQ–Orbitrap MS (Thermo Fisher Scientific, Waltham, MA, USA) [20], coupled with multiple data processing methods. Firstly, fragmental patterns of the tannins from *Sanguisorba* L. were analyzed based on the MS^1^ and MS^2^ information obtained by HPLC–LTQ–Orbitrap MS^2^ to acquire the cleavage pathways and diagnostic product ions (DPIs) for metabolite identification. Then, metabolite templates (known and identified metabolites) were summarized and established using the reported metabolic transforms of ingredients in the literature. Thirdly, compared with blank biosamples including urine, bile, plasma, feces, liver microsomes, liver cytosol, and intestinal microbiota, the metabolites from *Sanguisorba* L. in biosamples were identified using multiple metabolite templates, extracted ion chromatograms (EICs), and DPIs. Furthermore, we also used ChemDraw 14.0 (Thermo Fisher Scientific, Waltham, MA, USA) to calculated the Clog *p* values and the isomers were selected with different retention times. Often, the compound had a longer retention time when the Clog *p* value was greater in the reverse-phase chromatographic system.

### 2.2. Identification of Total Tannins Compounds Extracted from Sanguisorba Tannins

After concentration using HDP-400 macroporous resin column, the total tannins content in the final product was 65.5% and the main compounds extracted from *Sanguisorba* tannins were measured by using HPLC–LTQ–Orbitrap MS^2^. The main compounds detected are shown in Table 1, the total ion chromatograms of *Sanguisorba* tannins are shown in Figure 1, the summary analytical strategy diagram is shown in Figure 2 and their structures are shown in Figure 3. 

### 2.3. Identification of Metabolite of Sanguisorba Tannins In Vivo and In Vitro

According to the above analysis method, a large number of metabolites were identified in biological samples. All the detected metabolites of *Sanguisorba* tannins in vivo and in vitro are listed in Table 2, Table 3 and Table 4, respectively, the structures of the metabolites in vivo and in vitro are shown in Figure 4, and the metabolic pathways in vivo are shown in Figure 5.

### 2.4. Identification of Phase I Metabolites In Vivo

*1,6-Bis-O-(3,4,5-trihydroxybenzoyl)hexopyranose*, *Corilagin*, and their hydrolyzed metabolites. M1 eluted at 3.1 min and had a [M−H]^−^ ion at *m*/*z* 169.0148 (C_7_H_5_O_5_^−^, 3.54 ppm). The fragment ion at *m*/*z* 125.0241 (C_6_H_5_O_3_^−^) was generated by the loss of CO_2_ (44 Da) unit. M2 displayed a deprotonated molecular ion [M−H]^−^ at *m*/*z* 125.0257 (C_6_H_5_O_3_^−^, 2.05 ppm) at 6.7 min. The fragment ion at *m*/*z* 81.0345 (C_5_H_5_O^−^) was fragmented by the loss of CO_2_ (44 Da). Based on the mass spectra data and comparison with their reference substances, M1 and M2 were absolutely identified as gallic acid and pyrogallol, respectively [21].

M3 and P4 eluted at 28.1 min and 27.8 min, respectively, and had a [M−H]^−^ ion at *m*/*z* 300.9996 (C_14_H_5_O_8_^−^, 2.32 ppm). The fragment ions at *m*/*z* 283.1526 (C_14_H_3_O_7_^−^) and *m*/*z* 257.0825 (C_13_H_5_O_6_^−^) were generated by the loss of H_2_O (18 Da) unit and CO_2_ (44 Da) unit from the precursor ion, respectively. Based on the mass spectra data and comparison with their reference substances, M3 was absolutely identified as ellagic acid [22].

*Derived metabolites of hydrolysates*: M4 exhibited a protonated molecular ion [M−H]^−^ at *m*/*z* 275.0204 (C_13_H_7_O_7_^−^, 6.08 ppm) at 6.08 min. The fragment ion at *m*/*z* 257.0106 (C_13_H_5_O_6_^−^) was occurred via the loss of H_2_O (18 Da), suggesting that the loss was from the lactone. The fragment ions at *m*/*z* 229.0167 (C_12_H_5_O_5_^−^) through the elimination of the CO (28 Da) unit, which were consistent with those of M3. Based on the mass pattern, M4 was tentatively identified as a hydrolyzed product of M3, which indicated that M4 was urolithin M5 (3,4,8,9,10-pentahydroxy-urolithin) [22].

Metabolite M5 and M6 eluted at 25.2 min and 24.4 min, and are isomers that had [M−H]^−^ ion at *m*/*z* 259.0249 (C_13_H_7_O_6_^−^, 0.38 ppm) and 259.0251 (C_13_H_7_O_6_^−^, 1.16 ppm), respectively. The ion of M5 was at *m*/*z* 241.0176 (−18 Da) and its fragment ion was at *m*/*z* 213.0223 (−18 Da–28 Da); compared with the molecular ion of M4, there was a reduction by 16 Da. Based on the mass spectra data, it is speculated that M5 and M6 were formed by M4 dehydroxylation. The UV absorption spectrum of M6 shows two absorption peaks in the region of 240–400 nm, with an absorption peak of 349 nm, a second absorption peak of 263 nm, and a third absorption peak of 291 nm, similar to that of M4, and no significant redshift in band II. According to the literature [23], if no hydroxyl group is substituted on the para position (nine-position) of carbonyl group on urolith parent nucleus, there will be a redshift (displacement in the direction of longwave) in band I. Therefore, it is speculated that M6 retains the phenolic hydroxyl group in the para position of the carbonyl group, which is urolith D. The UV absorption spectrum of M5 showed three major absorption peaks in the region of 240–400 nm, respectively, at 251 nm, 275 nm, and 361 nm, with a significant redshift of band 2, compared with M4 (349 nm), suggesting that M5 is a nine-position hydroxyl removal metabolite of M4, which was named urolith M6.

M7 eluted at 28.4 min and had a [M−H]^−^ ion at *m*/*z* 243.0299 and compared with M6, its fragment ion is reduced by 16 Da. It is preliminarily determined to continue dehydroxylation from the metabolite M6 to obtain M7, and the UV absorption peak was 257 nm, 306 nm, and 345 nm, respectively. Consistent with the literature [24], identified M7 was urolithin C.

Metabolite M8 eluted at 30.4 min and had a [M−H]^−^ ion at *m*/*z* 227.0358, and its fragment ion decreased by 16 Da, compared with that of M7. It was tentatively identified that M8 was obtained from the metabolite M7 that continued to dehydroxyl, and the UV absorption peak was 238 nm, 306 nm, and 355 nm, respectively. Consistent with the literature [21], M8 was identified as urolithin A.

Metabolite M9 eluted at 38.1 min and had a [M−H]^−^ ion at *m*/*z* 211.0408, and its fragment ion decreased by 16 Da, compared with that of M8. It was tentatively identified that M9 was obtained from the metabolite M8 that continued to dehydroxyl, and the UV absorption peak was 238 nm, 306 nm, and 355 nm, respectively. Consistent with the literature [21], identified M9 was urolithin B.

The metabolite M10 eluted at 36.7 min and had a [M−H]^−^ ion at *m*/*z* 269.0424, and its fragment ions were 225.0454 (−16 Da–28 Da) and 197.0649 (−16 Da–28 Da), decreased by 32 Da (2O), compared with the molecular ion 257.0825 and 229.0167 of ellagic acid, speculating that M10 is a metabolite formed after the removal of two hydroxyl groups of ellagic acid, namely, 4, 9-dehydroxy-ellagic acid (Nasutin A) [21].

### 2.5. Identification of Phase II Metabolites In Vivo

M11 (retention time = 8.9 min) had a [M−H]^−^ ion at *m*/*z* 183.0294 (C_8_H_7_O_5_^−^, 3.19 ppm). The fragment ion at *m*/*z* 169.0049 (C_7_H_5_O_5_^−^) was fragmented by the loss of CH_2_ (14 Da), indicating that M7 was monomethylated conjugate. Moreover, the fragment ion *m*/*z* 125.0126 (C_6_H_5_O_3_^−^) was also observed. The fragmentation pattern of M11 is consistent with the literature [25]. Therefore, M11 was tentatively identified as methylation of M11. Metabolite M12 eluted at 4.2 min and had a [M−H]^−^ ion at *m*/*z* 197.0460 with 14 Da higher than that of M11; it was speculated that M12 was the metabolite M11 that continued to methylation. Hence, M12 was speculated to be gallic acid dimethyl.

M13 (retention time = 34.3 min) had a [M−H]^−^ ion at *m*/*z* 241.0513 (C_14_H_9_O_4_^−^, −2.9 ppm). The fragment ions at *m*/*z* 210.9041 and 196.9863 were detected; it is speculated that M13 was urolithin A methyl [25].

Metabolites M14, M15, M16, and M17 are glucuronic acid metabolites [25]. Their mass spectrometry all lost 176 Da, which proved to be phase II of combined with glucuronic acid metabolites. Metabolite 14 and 15 eluted at 26.6 min and 26.8 min and had [M−H]^−^ ions *m*/*z* 402.0678 and 419.0645, respectively. In their negative MS^2^ spectra, the fragment ions observed were *m*/*z* 227.0305 and *m*/*z* 243.0314, due to the loss of mass 176 Da corresponding to glucuronic acid. Therefore, metabolites 14 and 15 were urolithin A glucuronide and urolithin C glucuronide, respectively. Metabolite 16, 433.0787 [M−H]^–^ was obtained, fragment ions *m*/*z* 257.0485 and 243.0256 were obtained, after losing a methyl, the ionic peak of urolith C was obtained. Hence, M16 was identified as urolithin C methyl ether glucuronide. Metabolite M17 eluted at 29.4 min and had a [M−H]^−^ ion *m*/*z* 445.0792, and the fragment *m*/*z* 269.0463 (−176 Da) was obtained, speculating it was a glucuronic acid conjugate of 4,9-dehydroxy-ellagic acid (Nasutin A glucuronide).

Metabolite M18 eluted at 31.8 min and had a [M−H]^−^ ion at *m*/*z* 315.0140, and its fragment ion was 300.3491 (−15 Da), higher 15 Da (CH_3_) than ellagic acid, speculating that metabolite M18 was ellagic acid methyl ether. Metabolite M19 eluted at 25.8 min and had a [M−H]^−^ ion at *m*/*z* 491.0013, and its fragment ion was 315.0128 (−176 Da), decreasing 176 Da (C_6_H_9_O_6_) compared with M18, which proved to be phase II of combined with glucuronic acid metabolites; hence, Metabolite M19 was ellagic acid methyl ether glucuronide.

### 2.6. The Identification and Characterization of Metabolites In Vitro

Some metabolites in intestinal microbiota, liver microsomes, and liver cytosol can also be detected in vivo metabolism; moreover, new compounds were detected in vitro metabolism, proving that in vitro–in vivo metabolism is different. For example, metabolite N5, metabolite N8, and metabolite N9 were detected in intestinal microbiota.

### 2.7. Identification of Phase II Metabolites

Its mass spectrometry lost 176 Da, which proved to be phase II of combined with glucuronic acid metabolites. Metabolite N5, eluted at 9.6 min, had a [M−H]^−^ ion *m*/*z* 359.0264. In the negative MS spectra, the fragment ions observed were *m*/*z* 183.0205 and *m*/*z* 169.1322, due to the loss of mass 176 Da, corresponding to glucuronic acid. Therefore, metabolite N5 was gallic acid methyl ether glucuronide.

Metabolite N8 showed the HPLC profile with the retention time at 8.5 min and had the [M−H]^−^ ion at *m*/*z* 309.9918, and the fragment ions was 227.0374 (−80 Da), decreased by 80 Da (SO_3_) compared with the urolithin A, which indicated that N8 was sulfate conjugation metabolite.

N9 (retention time = 37.8 min) had a [M−H]^−^ ion at *m*/*z* 257.0458 (C_14_H_10_O_5_^−^, 0.78 ppm). The fragment ion at *m*/*z* 243.0276 (−14 Da) was obtained. Moreover, the fragment ion *m*/*z* 215.0385 (C_12_H_8_O_4_^−^) was also observed. Therefore, N9 was urolithin C methyl.

## 3. Discussion

### 3.1. Metabolic Pathways of Sanguisorba Tannins

The metabolism of *Sanguisorba* tannins in rats after oral administration in vivo (plasma, urine, bile, feces, and tissues) and in vitro (liver microsomes, liver cytosol, and intestinal microbiota) through incubation was elaborated in this study. Results are presented in what follows.

HPLC–LTQ–Orbitrap MS^2^ identified 19 metabolites in rats, including 4 metabolites from plasma, 9 metabolites from bile, 10 metabolites from urine, 5 metabolites from feces, 1 metabolite from kidney, 5 metabolites from liver microsomes and cytosol, and 9 metabolites from intestinal microbiota. According to the main components in the alcohol extract of *Sanguisorba* tannins contained and the structures of its metabolites, it can be inferred that after oral administration of tannins, it rapidly decreased in the blood, entered the liver, and expelled from bile, during which nine metabolites were detected. Figure 6 shows the proposed metabolic profiles of *Sanguisorba* tannins in rats, in liver microsomes, and intestinal microbiota, respectively. Gallic acid and ellagic acid were detected in liver microsomes and cytosol, and their conjugation of methyl was also detected, then absorbed into the blood and excrete from bile for the next metabolic reaction. M4–M10 were dehydroxylation metabolites of phase I, which were mainly detected in feces. Eventually, urolithin A is formed in feces. The phase II metabolites were urolithins of their derivatives and the combination with glucuronic acid, which were mainly detected in urine. Among them, urolithin A was also detected in intestinal microbiota; it is speculated that after oral *sanguisorba* tannins, the metabolites of phase I appeared first in intestinal microbiota, and a kind of smaller polarity such as urolithins was formed, then absorbed into the body; afterward, conjugation of phase II was formed, which became more polar and excreted from the urine. These results manifested that tannins mainly underwent reduction, hydrolysis, glucuronide conjugation, sulfate conjugation, and methylation changes. It should be noted that reduction reactions were the main metabolic steps. The conversion was further carried out by sulfation, followed by glucuronidation.

### 3.2. Comparison of Metabolites In Vitro and In Vivo

Metabolism research plays an important role in understanding the configuration of food and drugs and provides a basis for the study of the safety and toxicity of food and drugs [26]. The approach in vivo is quantitative and very effective in drug metabolism studies [27]. However, because of complex biomatrix, drug metabolites are often difficult to characterize in vivo [28]. Therefore, in vitro metabolic model is necessary to avoid the influence of other biomatrixes on metabolism in vivo. The identification of in vitro metabolites can supplement in vivo metabolites in complex biological samples. In vitro, culture methods are generally applicable to targeted studies and often predict the risk of real harm [29].

*Sanguisorba* tannins are the main tannins in *Sanguisorba*, play the main role in *Sanguisorba*, and have made a great contribution to the biological activity of *Sanguisorba*. However, research on its metabolic transport modes and pathways in the body is relatively weak, and the biotransformation of *Sanguisorba* tannins is mostly focused on hydrolyzable tannins. Research on the metabolism of condensed tannins in the body is relatively superficial. What is the relationship between the metabolic mechanism of *Sanguisorba* tannins in the body and the performance of its pharmacological activities? It needs further investigation.

## 4. Materials and Methods

### 4.1. Plant Materials

The dried *Sanguisorba* (FY2048) about 1 kg was collected at FEIYUBIO medicine market from Hubei Province, China, in June 2018, identified as *Sanguisorba* by Professor Zhang Hong of Wuhan University, ground to a fine powder in a grinder, passed through a 60-mesh (pore size in 0.25 mm) sieve and stored at 4 °C until analysis.

### 4.2. Chemicals and Materials

The authentic standards of ellagic acid (EA), pyrogallol, and gallic acid (GA) were purchased from the Chengdu MUST Bio-Technology CO. Ltd. (Chengdu, China), phosphate-buffered saline (PBS, pH = 7.4, 0.01 M), HPD resin was purchased from Tianjin Baowen Chemical Technology CO. Ltd. (Tianjin, China), anaerobic bags (AnaeroPouch Anaero, Mitsubishi Gas Chemical Company Inc., Tokyo, Japan), GAM broth medium was bought from Qingdao Haibo Biology CO. Ltd. (Qingdao, China), β-nicotinamide adenine dinucleotide phosphate (NADP), glucose-6-phosphate (G-6-P), glucose-6-phosphate dehydrogenase (G-6-PD), dithiothreitol, and S-adenosyl-L-methionine (SAM) were obtained from Sigma Chemical CO. (St. Louis, MO, USA). Acetonitrile, methanol, and formic acid were of HPLC-grade, which were purchased from Thermo Fisher Scientific CO. Ltd. (Waltham, MA, USA). Purified water was prepared by using Milli-Q System (Millipore, Billerica, MA, USA). All other analytical-grade reagents were provided by the Sinopharm Chemical Reagent CO. Ltd. (Shanghai, China).

### 4.3. Instrumentations and Investigation Conditions

Separation was carried out on a Welch Ultimate XB-C_18_ HPLC column (250 × 4.6 mm, 5 µm) at 35 °C. The mobile phase system was composed of acetonitrile (A) and 0.1% formic acid aqueous solution (B). The gradient program was used as follows: 0–20 min, 95% B; 20–25 min, 95–80% B; 25–30 min, 80–70% B; 30–45 min, 70–95% B. The flow rate was set at 0.80 mL/min. Two microliters supernatant samples were injected into LTQ–Orbitrap XL mass spectrometer system (Thermo Scientific, Bremen, Germany) for analysis.

The operating conditions of mass spectrometry were as follows: electrospray ionization (ESI) source was employed, and positive and negative ion mode were selected; sheath gas flow, 40 A.U.; auxiliary gas flow, 20 and 10 A.U.; source voltage 3.8 kV; capillary temperature of 300 °C; capillary voltage 25 and −35 V; tube lens, 110 and −110 V. The samples were analyzed using FT full scan with mass in the *m*/*z* 100–1500 range. All the raw data were processed using Xcalibur 3.0 software^19^ (Thermo Fisher Scientific, Waltham, MA, USA).

### 4.4. Sample Preparation and Pretreatment

#### 4.4.1. Extraction and Purification of the Total Tannins from *Sanguisorba*

According to the assay reported by [30], ultrasonic (Ningshang Ultrasonic Instrument CO. Ltd. Shanghai, China) assisted ethanol extraction of crude polyphenols was used. Preweighed amounts of *Sanguisorba* powder were placed into a volumetric flask (100 mL), soak dry ground *Sanguisorba* powder for 20 g with 70% ethanol solution at 20:1 solvent-to-sample ratio (*v*/*w*), and the extraction was placed in an ultrasonic cleaning bath at 40 °C for 1 h. Then, filtered through 0.22 μm filter paper to obtain the filtrate, while the residue was extracted again under the same extraction conditions, collected together with the filter, and concentrated by rotary vacuum evaporator (IKA-Werke-GmbH & CO., Staufen, Germany) to obtain the powders. Refer to the purification and separation methods [31] summarized in the previously published results of the research group. The HPD-400 macroporous resin column (Macklin biochemical CO. Ltd., Shanghai, China) was used for purification. Briefly, the crude polyphenolic extract was dissolved in sterile water to obtain the final concentration of 1 mg/Ml. Then, the concentrate was loaded into HPD-400 macroporous adsorption resin column (16 × 300 mm) at a flow rate of 2 mL/min. The final eluent was collected, concentrated by rotary vacuum evaporator, and freeze dried to obtain the tannic powder of *Sanguisorba* and kept sealed in the dark.

#### 4.4.2. Quantitation and Chemical Analysis of Total Tannins from *Sanguisorba*

After purified by HDP-400 macroporous resin column, the content of the total tannins in the final product was measured according to the spectrophotometric method from the Chinese Pharmacopoeia, 2010 edition. The total tannins were slightly modified according to the Folin–Ciocalteu method [32]. In brief, 0.2 mL of the extract (2 mg/mL) was mixed with 1 mL of the Folin–Ciocalteu’s reagent and 2 mL of 150 mg/mL Na_2_CO_3_. The 10 mL volumetric flasks were shaken and allowed to stand for 2 h at room temperature. The absorbance was measured with a spectrophotometer (Shimadzu CO., Tokyo, Japan) at 765 nm against a reagent blank and gallic acid was used as the standard with the contraction ranging from 2.909 to 6.545 μg/mL. All assays were run in three replicates. The total tannin content value was expressed as milligrams of gallic acid equivalents (GAE) per gram of dry weight (DW) (mg GAE/g DW). The method of determining nonadsorbed phenol content was the same as the determination of total phenol content, except that 0.6 g of casein was added to the sample. As a result, the total tannin content is total phenol content minus nonadsorbed phenol content, which is 65.5%.

### 4.5. Animal Experiments

The male Sprague Dawley (SD) rats, aged 6 weeks (200–220 g), were obtained from the Center of Experimental Animals of Medical College, Wuhan University. The rats were housed in the SPF breeding house of the animal experiment center of the people’s hospital of Wuhan university (temperature: 22–25 °C, relative humidity: 55–60%, light/dark cycle for 12 h) for seven days. All rats were fed on a standard pellet diet without tannins and water was freely available.

*Sanguisorba* tannins were suspended in saline. The 72 rats were randomly divided into 12 groups with 6 rats in each group (group 1, the blank plasma sample group; group 2, the experimental plasma sample group; group 3, the blank bile sample group; group 4, the experimental bile sample group; group 5, the blank urine and feces sample group; group 6, the experimental urine and feces sample group; group 7, the blank liver microsomes and cytosol sample group; group 8, the experimental liver microsomes and cytosol sample group; group 9, the blank intestinal microbiota sample group; group 10, the experimental intestinal microbiota sample group; group 11, the blank tissues sample group; group 12 the experimental tissues group). Before the experiment, the rats in the 12 groups fasted for 12 h with free access to water. The rats in vivo experiments group were orally administered *Sanguisorba* tannins at a single dose of 150 mg/kg (crude drug weight/rat weight). At the same time, saline was orally administered as blank control group rats. The rats in vitro experiments group, liver microsomes and cytosol, intestinal microbiota were incubated with *Sanguisorba* tannins (100 mg/mL, suspended in saline).

### 4.6. Sample Preparation

#### 4.6.1. Sample Collection and Pretreatment In Vivo

Plasma samples collection: After gavage, 200 μL of blood was obtained from the canthus in rats at 0, 1.5, 4, 10, 20, 40, 60, 90, 120, 180, and 240 min. The blood with three volumes of acetonitrile was centrifuged at 4500 rpm for 5 min, then the supernatant was centrifuged at 10,000 rpm/min for 15 min, and the supernatant was taken [33].

The urine and feces collection: a total of 12 rats were divided into two groups at random and housed in individual stainless steel metabolic cages designed for the separation and collection of urine and feces. Urine and feces were collected for 72 h post dosing. After measuring the volume of urine samples and dry weight of feces samples, an aliquot of 3 mL mixed urine was prepared using Grace Pure solid-phase extraction (SPE) C_18_ columns (Deerfield, IL, USA). Prior to sample preparation, SPE columns were conditioned using 5 mL methanol and subsequent 5 mL deionized water. Then, urine was loaded on the SPE cartridge, rinsed with 5 mL of deionized water, and eluted with 3 mL of methanol. The methanol eluent was collected and then evaporated to dryness at 40 °C under a gentle stream of nitrogen. Finally, the residues were redissolved using 100 μL 5% acetonitrile and vortex-mixed for 3 min. The 1 g of dried feces were dissolved with deionized water with ultrasonic processing for 60 min, after which the supernatant was collected after centrifuging at 3500 rpm/min for 10 min and concentrated [34].

The bile collection: The rats were orally administered drugs and then anesthetized with 2% pentasorbital sodium (45 mg/kg); the cannulas were surgically inserted into the bile duct to collect bile. An aliquot of 50 μL mixed bile samples were collected before dosing and during 0–4, 4–8, 8–12, 12–24, 24–36, and 36–48 h after dosing. The bile with three volumes of acetonitrile was centrifuged at 10,000 rpm/min for 5 min [35]. The study was carried out in compliance with the ARRIVE guidelines. (https://arriveguidelines.org, accessed on 13 July 2018).

The tissues collection: The rats were sacrificed 4 h after oral administration then anesthetized with 2% pentasorbital sodium (45 mg/kg), collected heart, liver, spleen, lung, kidney, stomach, and small intestine. All tissue samples were washed with saline to remove the blood and content, blotted on filter paper, weighed, and were added three volumes of saline for homogenate. Equal volume methanol was added to the homogenate and vortex-mixed for 1 min, then centrifuged at 10,000 rpm/min for 15 min, and the supernatant was collected. The organic phase was dried and extracted with ethyl acetate and centrifuged at 3000 rpm/min for 5 min, the ethyl acetate phase was extracted by the same method, and the organic phase was evaporated to dryness. Finally, the residues were redissolved by using 100 μL 5% acetonitrile and vortex-mixed for 3 min [36]. The study was carried out in compliance with the ARRIVE guidelines. (https://arriveguidelines.org, accessed on 13 July 2018).

All biological samples were stored at −80 °C and filtered through a 0.45 μm membrane filter until analysis.

#### 4.6.2. Sample Preparation and Treatment In Vitro

Preparation of liver microsomes and cytosol: Rat liver microsomes and cytosol were prepared by previous methods [37]. After anesthetized with 2% pentasorbital sodium (45 mg/kg), the liver of the rats was taken, rinsed with ice-cold normal saline, weighed, and homogenized in 0.1 mM phosphate buffer (pH7.4) containing 0.25 M sucrose. The homogenate was centrifuged at 4 °C for 30 min (10,000× *g*), and the supernatant was centrifuged at 4 °C for 60 min (105,000× *g*). The supernatant was liver cytosol and the precipitation was liver microsomes. The protein content was determined by the BCA method [37]. The study was carried out in compliance with the ARRIVE guidelines. (https://arriveguidelines.org, accessed on 13 July 2018).

The rat liver microsomes incubation was prepared as follows: 100 μM *Sanguisorba* tannins, 1.3 mM NADP, 3.3 mM G-6-P, 0.4 U/mL G-6-P-D, 3.3 mM magnesium chloride. The incubation system was achieved with Tris/KCl buffer (pH 7.4) and its volume was 250 μL. After preincubation for 5 min at 37 °C, add NADPH mixture to start the reaction, which including 1.3 mM NADP, 3.3 mM G-6-P, 0.4 U/mL G-6-P-D, and 3.3 mM magnesium chloride. Incubated at 37 °C for two hours and add equal volume acetonitrile to terminated the reaction. The incubation was centrifuged at 10,000× *g* for 10 min, the supernatant evaporated to dryness at 40 °C under a gentle stream of nitrogen. The residues were dissolved with 80 μL acetonitrile, and 20 μL aliquot was analyzed by HPLC–LTQ–Orbitrap MS^2^.

The optimal incubation conditions (200 μL reaction volume) for rat liver cytosol were as follows: 100 μM *Sanguisorba* tannins, 5 mM magnesium chloride, 1 mM dithiothreitol, and 300 μM SAM. The incubation system was achieved with Tris/KCl buffer (pH 7.4), and its volume was 200 μL. After preincubation for 5 min at 37 °C, add rat liver cytosol to start the reaction. Incubated at 37 °C for two hours and add equal volume acetonitrile to stop the reaction. The incubation was centrifuged at 10,000× *g* for 10 min, the supernatant evaporated to dryness at 40 °C under a gentle stream of nitrogen. The residues were dissolved in an 80 μL acetonitrile, and 30 μL aliquot was analyzed by HPLC–LTQ–Orbitrap MS^2^.

Preparation of rat intestinal microbiota: Anaerobic culture medium was prepared as follows: K_2_HPO_4_ (37.5 mL, 0.78%), solution A (37.5 mL, 0.47% KH_2_PO_4_, 1.18% NaCl, 1.2% (NH_4_)_2_SO_4_, 0.12% CaCl_2_, and 0.25% MgSO_4_·H_2_O), Na_2_CO_3_ (50 mL, 8%), L-cysteine (0.5 g), L-ascorbic acid (26 mL, 25%), eurythrol (1 g), tryptone (1 g), and nutrient agar (1 g) were mixed together and diluted with distilled water to 1000 mL. The solution was then adjusted to pH 7.5–8.0 with 2 M HCl [38]. Fresh feces collected from Sprague Dawley rats were immediately homogenized in normal saline solution at a ratio of 1 g to 4 mL. The homogenate was filtered, and the filtrate of fresh feces (10 mL) was added to an anaerobic culture medium (90 mL) to obtain an intestinal microbiota cultural solution.

The rat intestinal microbiota incubation was prepared as follows: 100 μM *Sanguisorba* tannin was added to the intestinal microbiota cultural solution (200 μL). After incubation for 0, 2, 4, 6, 12, 24, 36, 48, and 72 h at 37 °C in an anaerobic environment, equal volume acetonitrile was added to terminate the reaction. The incubation was centrifuged at 10,000× *g* for 10 min, the supernatant evaporated to dryness at 40 °C under a gentle stream of nitrogen. The residues were dissolved in an 80 μL acetonitrile, and 30 μL aliquot was analyzed by HPLC–LTQ–Orbitrap MS^2^.

## 5. Conclusions

In conclusion, using HPLC–LTQ–Orbitrap MS^2^ for analysis, high-resolution mass spectrometry solves the deficiencies of the above methods by virtue of its ultra-high resolution and accurate mass function, especially for multicomponent screening. Additionally, a variety of post-acquisition data mining tools were used. This method has good sensitivity and selectivity and is suitable for qualitative analysis for screening purposes; the metabolites in vivo and in vitro of *Sanguisorba* tannins were successfully identified. A total of 19 metabolites in vivo and 14 metabolites in vitro were recognized in this study; no active drug, ellagic acid, or other metabolites were detected in rat plasma; after the treatment of the tissues, no active drug or metabolites were detected in them, but ellagic acid was detected in the liver and methyl ether glucuronide in the kidney. Moreover, seven metabolites in vitro were also found in vivo; N5, N8, and N9 were detected in vitro but not found in vivo, which indicated metabolic mechanism between in vivo and in vitro was different. The basic metabolic changes that occurred in rats were reduction, which provides a basis for other metabolic reactions such as glucuronide conjugation, sulfate conjugation, and methylation. Finally, the metabolic profile of *Sanguisorba* tannins was reviewed. High-performance liquid chromatography–mass spectrometry, combined with various data processing techniques, provides valuable information for the identification of metabolites.

*Sanguisorba* tannins were the main content in *Sanguisorba*, which have drawn growing attention due to the prominent biological activities of its metabolites. For example, urolithin A, C, and D can reduce the accumulation of triglyceride, reducing the risk of animation atherosclerosis [39]; urolithiasis A and its sulfate conjugation may protect against breast cancer [40]; urolithin A has anti-inflammatory and antioxidant effects of [41]; urolithin A glucuronide and its aglycone urolithin A ameliorate TNF-alpha-induced inflammation [42], which made a great contribution to biological activity. We hope that the strategies developed in this study and the results achieved in this study will be helpful for future studies on the effectiveness and safety of tannins in *Sanguisorba*. Based on the analysis of its metabolites, the tannin’s metabolism and transport pathway can be speculated in vivo, which further elucidated the mechanism of its role.

However, there are many restrictions. The components of traditional Chinese medicine are complex, not a single ingredient, and therefore, it was not possible to trace the metabolic pathways of individual components in the body. Only by combining in vivo and in vitro, by detecting the metabolites in different organs, we can analyze and detect the metabolic pathways in the body. Further study on the activity of its metabolites will be carried out in the following part to provide the basis for the development and utilization of new drugs.

## Figures and Tables

**Figure 1 molecules-26-04053-f001:**
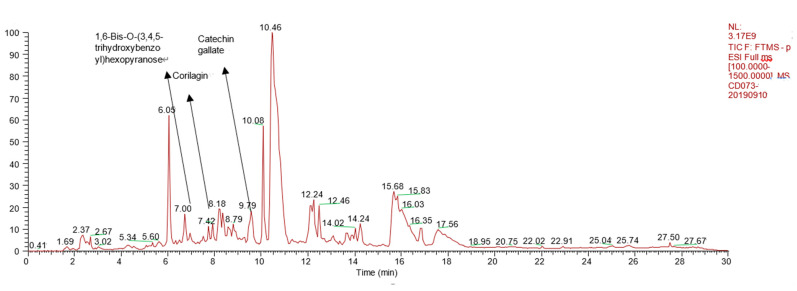
Total ion chromatograms of *Sanguisorba* tannins.

**Figure 2 molecules-26-04053-f002:**
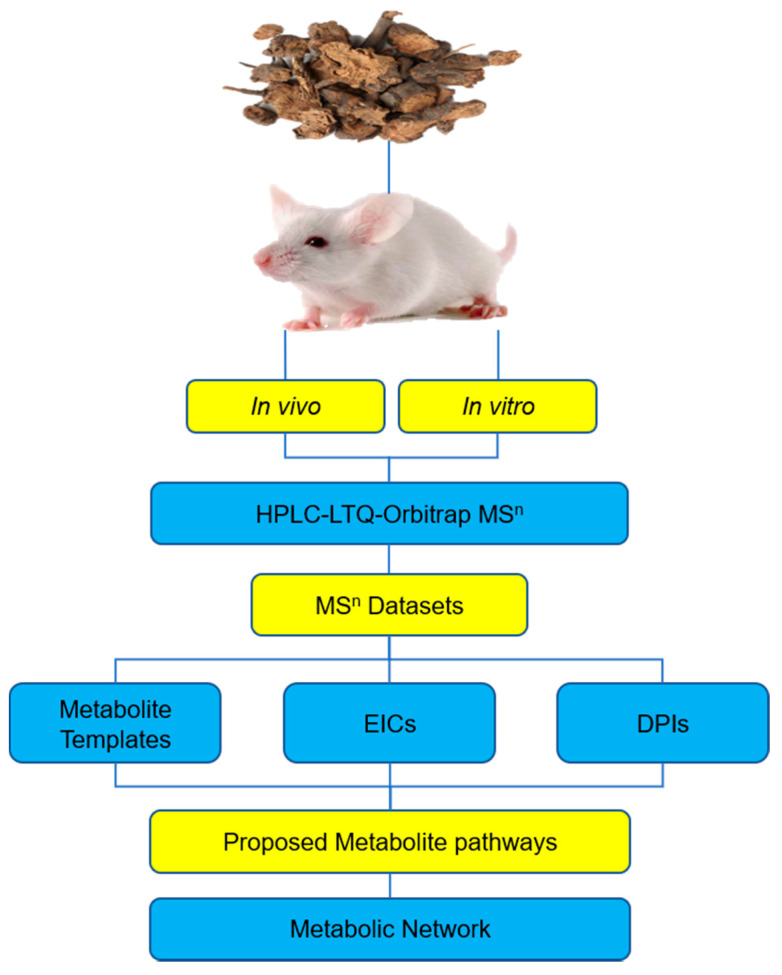
The summary analytical strategy diagram for the detection and identification of *Sanguisorba* tannins: EICs—extracted ion chromatograms, DPIs—diagnostic product ions.

**Figure 3 molecules-26-04053-f003:**
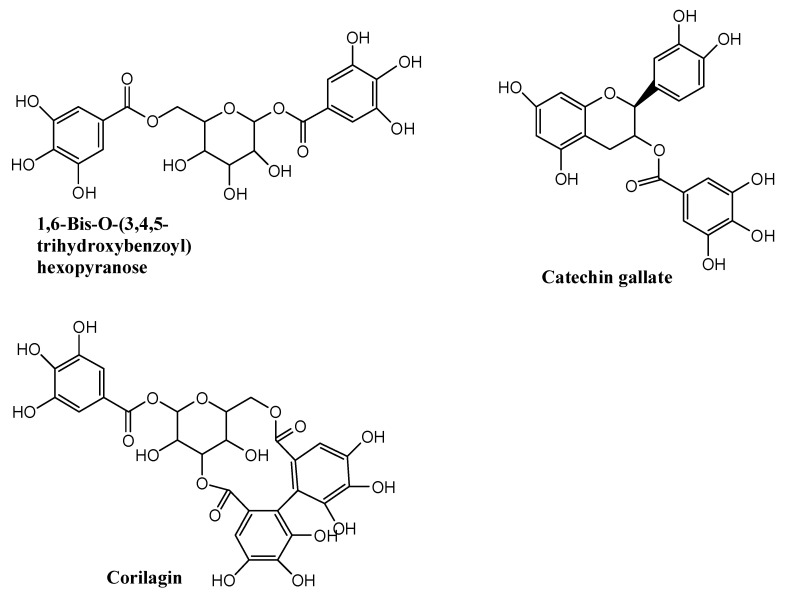
The summary contents structure in *Sanguisorba* tannins.

**Figure 4 molecules-26-04053-f004:**
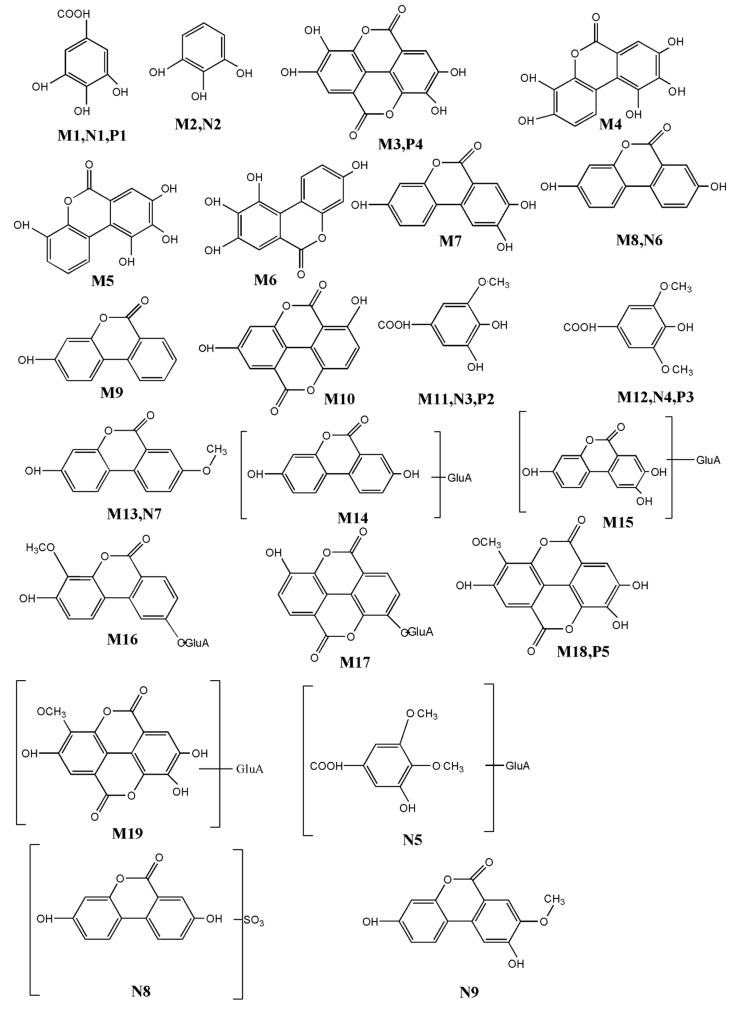
Metabolic profile and the proposed metabolic pathways of *Sanguisorba* tannins in vivo and in vitro.

**Figure 5 molecules-26-04053-f005:**
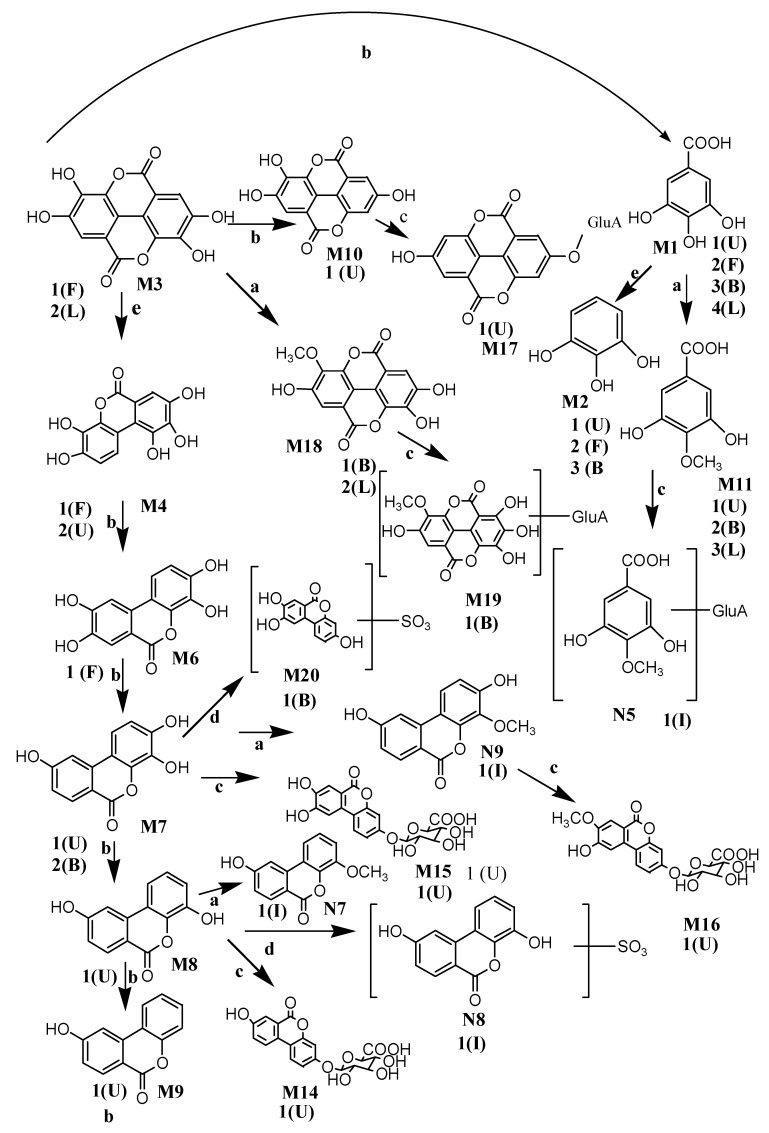
The summary metabolites structure in vivo and in vitro: a—methylation, b—dihydroxylation, c—glucuronidation conjugation, d—sulfation conjugation, e—decarbonylation, U—urine, P—plasma, F—feces, I—intestinal flora, L—liver microsomes and liver cytosol.

**Figure 6 molecules-26-04053-f006:**
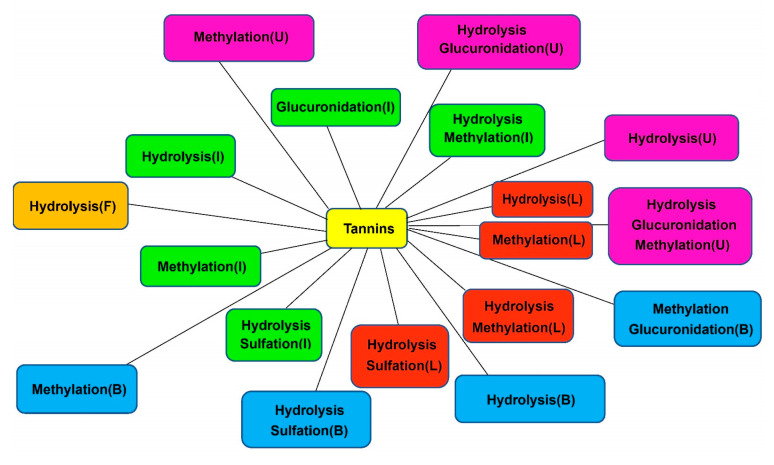
The metabolic networks of *Sanguisorba* tannins: U—urine, F—feces, B—bile, I—intestinal flora, L—liver microsomes and liver cytosol.

**Table 1 molecules-26-04053-t001:** Tannins profile in *Sanguisorba* ethanolic extract.

NO.	Name	Formula	tR (min)	MS^1^	MS^2^	mzCloud Best Match
1	1,6-Bis-O-(3,4,5-trihydroxybenzo yl)hexopyranose	C_20_H_20_O_14_	7.00	484.1111	153.0137	77.9
2	Catechin gallate	C_22_H_18_O_10_	9.79	442.0894	151.0387 139.0380 123.0440	79.1
3	Corilagin	C_27_H_22_O_18_	7.42	634.4729	463.0524 300.9995	89.3

**Table 2 molecules-26-04053-t002:** Summary of phase I and phase Ii metabolites of *Sanguisorba* tannins in SD rats’ plasma, bile, urine, feces, and tissues samples: ^I^—phase I metabolites, ^II^—phase II metabolites.

NO.	tR (min)	MS^1^	Diff(ppm)	MS^2^	Formula	Compound	Occurrence
M1 ^I^	3.1	169.0148	3.54	125.0241	C_7_H_6_O_5_	Gallic acid	Urine Feces Bile plasma
M2 ^I^	6.7	125.0257	2.05	81.0345	C_6_H_6_O_3_	Pyrogallol	Urine Feces Bile Plasma
M3 ^I^	28.1	300.9996	2.32	283.1526 257.0825	C_15_H_6_O_8_	Ellagic acid	Feces
M4 ^I^	24.1	275.0204	6.08	257.0106 229.0167	C_13_H_8_O_7_	Urolithin M5 (3,4,8,9,10-pentahydroxy-urolithin)	Urine Feces
M5 ^I^	25.2	259.0249	0.38	241.0176 213.0223	C_13_H_8_O_6_	Urolithin M6	Feces
M6 ^I^	24.4	259.0251	1.16	241.0171 213.0225	C_13_H_8_O_6_	Urolithin D	Feces
M7 ^I^	28.4	243.0299	0.82	215.0367 187.0426	C_13_H_8_O_5_	Urolithin C	Urine Bile
M8 ^I^	30.4	227.0358	4.40	199.0346	C_13_H_8_O_4_	Urolithin A	Urine
M9 ^I^	38.1	211.0408	3.79	167.0502 139.0556	C_13_H_8_O_3_	Urolithin B	Urine
M10 ^I^	36.7	269.0424	−1.12	224.0454 197.0649	C_14_H_6_O_6_	Nasutin A (4,9-dehydroxy-ellagic acid)	Urine
M11 ^II^	8.9	183.0294	2.73	169.0049 125.0126	C_8_H_8_O_5_	Gallic acid methyl	Urine Bile
M12 ^II^	4.2	197.0460	0.51	183.0854 169.0146 125.0196	C_9_H_10_O_5_	Gallic acid dimethyl	Bile
M13 ^II^	34.3	241.0513	−2.9	210.9041 196. 9863	C_14_H_10_O_4_	Urolithin A methyl	Bile
M14 ^II^	26.6	402.0678	7.89	227.0305 227.1005 175.2006	C_19_H_16_O_10_	Urolithin A glucuronide	Urine
M15 ^II^	26.8	419.0645	1.53	243.0314 214.9997 187.0426	C_19_H_16_O_11_	Urolithin C glucuronide	Urine
M16 ^II^	26.5	433.0787	5.07	257.0485 243.0256	C_20_H_18_O_11_	Urolithin C methyl ether glucuronide	Urine
M17 ^II^	29.4	445.0792	2.47	269.0463	C_21_H_18_O_11_	Nasutin A glucuronide	Urine
M18 ^II^	31.8	315.0140	−9.52	300.9991	C_15_H_8_O_8_	Ellagic acid methyl ether	Bile
M19 ^II^	25.8	491.0013	0. 26	315.0128 299.7270	C_21_H_16_O_14_	Ellagic acid methyl ether glucuronide	Bile Kidney

**Table 3 molecules-26-04053-t003:** Summary of phase I and phase II metabolites of *Sanguisorba* tannins in rat intestinal microbiota: ^I^—phase I metabolites, ^II^—phase II metabolites.

NO.	tR (min)	MS^1^	Diff(ppm)	MS^2^	Formula	Compound
N1 ^I^	2.2	169.0115	15.97	125.0225	C_7_H_6_O_5_	Gallic acid
N2 ^I^	5.8	125.3446	2.75	81.1342	C_6_H_6_O_3_	Pyrogallol
N3 ^II^	8.4	183.0298	0.54	169.0052 125.0177	C_8_H_8_O_5_	Gallic acid methyl
N4 ^II^	4.1	197.0466	0.54	169.0118 125.0232	C_9_H_10_O_5_	Gallic acid dimethyl
N5 ^II^	9.6	359.0264	1.86	183.0205 169.1322	C_14_H_16_O_11_	Gallic acid methyl ether glucuronide
N6 ^I^	32.3	227.0312	1.58	199.0344	C_13_H_8_O_4_	Urolithin A
N7 ^II^	36.6	241.0531	10.4	210.9047 196.9068	C_14_H_10_O_4_	Urolithin A methyl
N8 ^II^	8.5	309.9918	2.83	227.0374	C_13_H_7_O_7_S	Urolithin A sulfate
N9 ^II^	37.8	257.0458	0.78	243.0276 215.0385	C_14_H_10_O_5_	Urolithin C methyl

**Table 4 molecules-26-04053-t004:** Summary of phase I and phase II metabolites of *Sanguisorba* tannins in rat liver microsomes and liver cytosol: ^I^—phase I metabolites, ^II^—phase II metabolites.

NO.	tR(min)	MS^1^	Diff(ppm)	MS^2^	Formula	Compound
P1 ^I^	2.7	169.0115	2.96	125.4925	C_7_H_6_O_5_	Gallic acid
P2 ^II^	8.4	183.0292	−3.82	125.0236	C_8_H_8_O_5_	Gallic acid methyl
P3 ^II^	5.3	197.0414	0.51	125.0788	C_9_H_10_O_5_	Gallic acid dimethyl
P4 ^I^	27.8	300.9996	2.32	283.9996 257.0067 229.0102 185.0274	C_15_H_6_O_8_	Ellagic acid
P5 ^II^	30.8	315.0148	3.25	299.9936	C_15_H_8_O_8_	Ellagic acid methyl ether

## Data Availability

The data presented in this study are available in this article.

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
