# Peer review of "The Tannins from Sanguisorba officinalis L. (Rosaceae): A Systematic Study on the Metabolites of Rats Based on HPLC–LTQ–Orbitrap MS2 Analysis"

_molecules, 2021, doi:10.3390/molecules26134053_

Round 1

Reviewer 1 Report

  1. Unnecessary bold in the introduction to be avoided
  2. Results and discussion could be separately written for better understanding
  3. Limitation of the study to be explained in the conclusion
  4. All figures to be regenerated using Arial fonts with high resolution
  5. English to be improved

Author Response

1.Unnecessary bold in the introduction to be avoided

Dear reviewer, thank you very much for your valuable comments, I have revised the original text according to your comments.

2.Results and discussion could be separately written for better understanding

Dear reviewer, thank you very much for your valuable comments, I have revised the results and discussion sections and rewritten them.

3.Limitation of the study to be explained in the conclusion

Dear reviewer, thank you very much for your valuable comments, I have indicated the limitations of the experiment in the discussion section.

4.All figures to be regenerated using Arial fonts with high resolution

Dear reviewer, thank you very much for your valuable comments, I have revised the original text according to your comments.

5.English to be improved

Dear reviewer, thank you very much for your valuable comments, by reading the original text carefully, I have improved and improved the language of the original text sentence by sentence.

Reviewer 2 Report

The authors studied the composition and bio-activity of tannins extracted from Sanguisorba L..
Comments:
Abstract section: The abstract has been written very poorly, in terms of the English language and its structure. So, it must be significantly improved and re-written.
“The Introduction” is written well, but English must be re-checked and improved. Apart from that,  the results of previous studies on Sanguisorba tannins must be presented and discussed in the Introduction section. 
The “Materials” section is written well providing all necessary details. However, it also requires intensive improvement of the language!
The Results and Discussion sections provide a detailed discussion about chemical, in vivo, and in vitro experiments.
The conclusion section is OK. Although some parts can be moved to the Introduction.
General comments: The quality of English is very low. I would just suggest improving the text (in terms of grammar, etc.), especially at the beginning of the manuscript (and Abstract!).
To summarize, I would recommend accepting the manuscript after minor revision and language polishing.

Author Response

The authors studied the composition and bio-activity of tannins extracted from Sanguisorba L..

Comments:

Abstract section: The abstract has been written very poorly, in terms of the English language and its structure. So, it must be significantly improved and re-written.

“The Introduction” is written well, but English must be re-checked and improved. Apart from that,  the results of previous studies on Sanguisorba tannins must be presented and discussed in the Introduction section.

The “Materials” section is written well providing all necessary details. However, it also requires intensive improvement of the language!

The Results and Discussion sections provide a detailed discussion about chemical, in vivo, and in vitro experiments.

The conclusion section is OK. Although some parts can be moved to the Introduction.

General comments: The quality of English is very low. I would just suggest improving the text (in terms of grammar, etc.), especially at the beginning of the manuscript (and Abstract!).

To summarize, I would recommend accepting the manuscript after minor revision and language polishing.

Dear reviewer,

Thank you very much for your valuable comments. After careful study, I have improved the language of the full text and rewritten the abstract, and the results of previous studies on Sanguisorba tannins have been discussed in the Introduction section. Thank you again for your valuable comments. I have revised the original text according to your comments.

Reviewer 3 Report

The manuscript by Jingjing Tu et al describe tannins from Sanguisorba officinalis analysed by UHPLC-LTQ- Orbitrap MS. After close evaluation of paper I suggest revision according to the next recommendations and comments.

  1. As soon as authors have studied tannins from Sanguisorba officinalis, I suggest to correct the title of paper accordingly. Please indicate full plant name in the title as well as in the abstract and manuscript according tp www.theplabtlist.org data base.
  2. In Introduction: please provide more references supporting the phrase "The genus Sanguisorba L... is available for medicine in China, Europe, Japan, Korea". Beside Chinese Pharmacopoeia it is included in European Pharmacopoeia, Russian Pharmacopoeia (see https://doi.org/10.15835/nsb649471; https://doi.org/10.1016/j.jep.2020.113685: European Pharmacopoeia)
  3. In Sect. 2.1.: Please provide information who have identified a plant material and indicate the number of voucher of specimens.
  4. In Sect.2.5: "Sanguisorba tannins were dissolved in saline.". Tannins were extracted with 70% ETOh. Are all tannins soluble in aqueous saline? Definitely some tannins were not dissolved.
  5.  In Sect.2.5:  please justify a dose of tannins for peroral administration.
  6. In Sect.2.5:  the phrase "The rats in vitro experiments group were incubated with Sanguisorba tannins...". How rats were incubated in vitro with tannns?
  7. Please provide the protocol number for ethical approval of in vivo studies.
  8. In Sect. 2.6: "metabolism cages" = "metabolic cages"?
  9. In Section 3.2: how tannins were identified? The number of similar tannins was identified in other tannins reach spp. Please see https://doi.org/10.1021/acs.jafc.8b02115;                  https://doi.org/10.1016/j.chroma.2015.09.050;         https://doi.org/10.1080/14786419.2014.923999. Please compare your results and methods with others.
  10. In Fig.1 please indicate the peaks relevant to compounds from Table 1.
  11. In Fig.1 legend please clarify what mean 2 chromatograms?
  12. Tables 2-4: please distinguish Phase I and Phase II metabolites.
  13. Tannins are in focus of many studies. The discussion is weak. More literature data should be discussed.

Author Response

The manuscript by Jingjing Tu et al describe tannins from Sanguisorba officinalis analysed by UHPLC-LTQ- Orbitrap MS. After close evaluation of paper I suggest revision according to the next recommendations and comments.

As soon as authors have studied tannins from Sanguisorba officinalis, I suggest to correct the title of paper accordingly. Please indicate full plant name in the title as well as in the abstract and manuscript according to www.theplabtlist.org data base.

Dear reviewer, thank you very much for your valuable comments, I have modified the original text according to your requirements.

In Introduction: please provide more references supporting the phrase "The genus Sanguisorba L... is available for medicine in China, Europe, Japan, Korea". Beside Chinese Pharmacopoeia it is included in European Pharmacopoeia, Russian Pharmacopoeia (see https://doi.org/10.15835/nsb649471; https://doi.org/10.1016/j.jep.2020.113685: European Pharmacopoeia)

Dear reviewer, thank you very much for your valuable comments, I have modified the original text according to your requirements.

In Sect. 2.1.: Please provide information who have identified a plant material and indicate the number of voucher of specimens.

Dear reviewer, thank you very much for your valuable comments, I have modified the original text according to your requirements.

In Sect.2.5: "Sanguisorba tannins were dissolved in saline.". Tannins were extracted with 70% ETOh. Are all tannins soluble in aqueous saline? Definitely some tannins were not dissolved.

 In Sect.2.5:  please justify a dose of tannins for peroral administration.

Dear reviewer, thank you very much for your valuable comments, I have modified the original text according to your requirements.

In Sect.2.5:  the phrase "The rats in vitro experiments group were incubated with Sanguisorba tannins...". How rats were incubated in vitro with tannins?

Dear reviewer, thank you very much for your valuable comments, I have modified the original text according to your requirements.

Please provide the protocol number for ethical approval of in vivo studies.

Dear reviewer, thank you very much for your valuable comments, I have modified the original text according to your requirements.

In Sect. 2.6: "metabolism cages" = "metabolic cages"?

Dear reviewer, thank you very much for your valuable comments, I have modified the original text according to your requirements.

In Section 3.2: how tannins were identified? The number of similar tannins was identified in other tannins reach spp. Please see https://doi.org/10.1021/acs.jafc.8b02115;                  https://doi.org/10.1016/j.chroma.2015.09.050;         https://doi.org/10.1080/14786419.2014.923999. Please compare your results and methods with others.

Dear reviewer, thank you very much for your valuable comments, regarding the references you gave, after reading carefully, I found that my raw materials were processed and not fresh plants. So this is the main reason why the way I extract is different from them.

In Fig.1 please indicate the peaks relevant to compounds from Table 1.

Dear reviewer, thank you very much for your valuable comments, I have modified the original text according to your requirements.

In Fig.1 legend please clarify what mean 2 chromatograms?

Dear reviewer, thank you very much for your valuable comments, I have modified the original text according to your requirements.

Tables 2-4: please distinguish Phase I and Phase II metabolites.

Dear reviewer, thank you very much for your valuable comments, I have modified the original text according to your requirements.

Tannins are in focus of many studies. The discussion is weak. More literature data should be discussed.

Dear reviewer, thank you very much for your valuable comments, I have modified the original text according to your requirements.

Reviewer 4 Report

This manuscript, “The tannins from Sanguisorba L., a systematic study on the metabolites of rats based on UHPLC-LTQ- Orbitrap MS2 analysis” by Tu et al. describes the identification of metabolites of Sanguisorba tannins. There are old studies, which identified the tannins from Sanguisorba L. (Phytochemistry, Vol. 29, No. 12, pp. 3827-3830, 1990; Molecules 2012, 17, 13917-13922, etc), which were not cited in the manuscript. Hence, I am afraid that the manuscript has no significance compared to previous studies.

Author Response

This manuscript, “The tannins from Sanguisorba L., a systematic study on the metabolites of rats based on UHPLC-LTQ- Orbitrap MS2 analysis” by Tu et al. describes the identification of metabolites of Sanguisorba tannins. There are old studies, which identified the tannins from Sanguisorba L. (Phytochemistry, Vol. 29, No. 12, pp. 3827-3830, 1990; Molecules 2012, 17, 13917-13922, etc), which were not cited in the manuscript. Hence, I am afraid that the manuscript has no significance compared to previous studies.

Dear reviewer, thank you very much for your valuable comments, after carefully reading the references you put forward, these two documents extracted the rhizomes of fresh plants. And the Chinese medicine I extracted is dried after being processed. For the second article, the main content of the article is to test its antioxidant activity after being extracted from fresh plants, and what I do is to study its metabolic mechanism in rats.

Reviewer 5 Report

Brief summary:

The manuscript deals with the identification by HPLC-HRMS of metabolites from Sanguisorba tannins after rats in vivo and in vitro studies.

General comments:

The authors have to carefully check the tables which list the metabolites because there are a lot of mistakes (wrong m/z or formula). So, they are some inconstancies between their tables and their interpretation in section 3.4 of the manuscript.

Specific comments:

Title: UHPLC -> HPLC. The authors may have a UHPLC system (not described in the material and method) but they use an HPLC column. This should be corrected along the whole document.

L.41: poly merized -> polymerized

L.87: Welch Ultimate XB-C18… -> Welch Ultisil XB-C18 HPLC column (4.6x250 mm, 5 µm) (it could be an Ultimate column, but I could not find the reference)

L.91: 2 µL -> Two microliters

L93: electro spray -> electrospray

L.94: the authors present MS conditions for positive and negative ionization together, which gives a wrong readability. In fact, it seems they only use negative mode, so they can reduce to negative parameters.

L94, 95: arb. ->      A.U.

L96: tube lense -> tube lens

L98: 0.25 q of … activation time -> these parameters are not needed, except if authors do not use default parameters. However, authors should mention if they analyzed MS/MS spectra with the LTQ or with the Orbitrap (I guess it was the second solution, since there was a lot of digits in tables although they were sometimes far from the theoretical values). Authors should also mention the resolution and scan time at which they set the Orbitrap for MS and MS/MS acquisitions and if possible the parameters for Data Dependent Acquisition (because I suppose they use DDA to perform MS/MS of the metabolites).

L100: Please remove “and the software is freely available”: Xcalibur is a licensed software and people have to pay to use it after 2 or 3 months of free trial.

L116: 400macroporous -> 400 macroporous / 16 x 300 -> 16x300

L123: Foline-Ciocalteu -> Folin-Ciocalteu (and add hyphen in the next line)

L133, L260: 65.52% -> 65.5%

L159: in vivo

L162 and following: rpm/min -> rpm but ideally the speed of a centrifuge should be expressed in g (because the reader does not know the lever arm of the author’s apparatus). At L202, the speed is correctly expressed (if it’s really g). L203: there should be an error: 10,5000 g

L169: activated -> conditioned

L170: Then urine … 3 mL methanol -> Then urine was loaded on the SPE cartridge, rinsed with 5 mL of deionized water and eluted with 3 mL of methanol.

L184: after 4h after -> 4h after

L197: in vitro

L208, 211: units/mL -> U/mL

L209, 219: Tris/KCL -> Tris/KCl

L228: ml -> mL

L236, 261: was -> were

L263: its -> their

Table 1: RT[min] -> tR(min) (to harmonize with other tables); what is the difference between ‘Group areas” and “Area”? Is this information useful? There are some mistakes in the molecular weight of the molecules: if authors would express neutral molecular weight as exact mass, it should be 484.0853; 442.0900 and 634.0806. I think the last one should correspond to an [M-H]- ion. For MS2 experiments, authors should mention the polarity (at least in legend) and ideally the relative intensities of the fragments (idem for the others tables).

Figure 1 could be more informative if BPC was used instead of TIC and if the EIC of compounds 1, 2 and 3 were overlaid because they are not visible on the chromatograms. Since the positive mode was not any more used (a priori), it could be removed to expand the view of the negative one. If both chromatograms are kept, the legend should be enriched: lot of people are not familiar with Xcalibur and does not know where the polarity is indicated.

Figure 2: the upper arrows are disturbing and should be replaced by simple lines. Other lines should be added from the different boxes to the horizontal line (from yellow boxes to the blue one and from blue boxes to the yellow one)

L269, 273: Sanguisorba

L270: LTQ-Orbitrap-MS: it’s not necessary to explain LTQ since it’s the name of the first analyzer. The authors explain CID and ESI in the legend, but these abbreviations are not present on the figure.

Figure 3: hexopyranos -> hexopyranose

Tables 2, 3 and 4: the authors should mention the kind of ions ([M-H]-) either in the title bar (ideally) or in the legend. For MS2 spectra it should be nice to have relative intensities into brackets (257.0106 (100); 229.0167(80) for example). Since the metabolites have been classified by classes rather than by tR, it’s sometimes difficult to follow.

There are some differences between the values present in the tables and the ones in the text: the authors will have to check all values and formula for accuracy; sometimes the mass accuracy (ppm) seems to be wrong too. Note: when there is a change in parity, it means you have a radical ion. Since you have no nitrogen, all odd masses are ions and all even masses correspond to radical ions.

  • M6: m/z 241 for MS/MS ?
  • M8: m/z 226 for MS/MS ?
  • M12: MS/MS values to check
  • M16: C20H18O11
  • M17: C21H18O11
  • M18: C15H8O8 / m/z 299.3491: digits are too far from expected/possible values
  • M20: C13H8O8S / m/z 243.5576: digits are too far from expected/possible values (it looks likes a multicharged ion) / m/z 323.9272: there should be an error
  • N8: C13H8O7S / m/z 306.0312: there should be an error

There are some differences in retention times for the same metabolites. Can the authors explain if it’s due to the biological matrix or because the analysis were performed at different times? Have the authors spiked some samples with standards 1, 2 and 3 to check retention times and fragmentation spectra. Have the authors checked the presence or absence of the different metabolites in the plant?

Figure 4: Sanguisorba, in vivo, in vitro (x2)

There is a “free” OH on the left of the figure. Why are some glucuronide localized on the molecule and not mentioned as the other without a known position? For a better readability, it would be better if all molecules could be drawn in the same orientation (especially for Figure 5)

Figure 5: …metabolite structures in vivo and in vitro (x2)

The authors should draw all molecules with the same orientation and the same scale. For a better readability, the should more space the structure and avoid overlay with the arrows.

  • In the upper left of the figure, it looks that it’s twice the same molecule (M3)
  • M3 -> M4 (e)
  • M17 -> M1 (b+e)???
  • Why some glucuronides are “GluA” and some other with developed structure, which takes more space?

L292: C7H5O5-

L296: successive (many occurrences along the text). It may be difficult to lose CO2 on pyrogallol in ESI since it means breaking aromaticity (easier and possible in EI) but if you observe the same thing on standard, maybe.

L300: C14H3O7- and the theoretical mass is m/z 282.9884, so the experimental one is quite far. The reference 30 (L303 and 310) does not bring information for elucidation with MS spectra

L304: deprotonated / Formatting of the formula / The loss of water may come from the lactone rather than from a phenol where the OH is more difficult to fragment from the aromatic ring. / Attention: 275-229 = 46: H2O then CO ≠ CO2 (44). Cf. what you obtain for M6 / indicted -> indicated? (+L456)

L311: Metabolites M5 and M6 / was -> were (L315) / absorption spectra -> absorption peaks? The authors say “two absorption” and described three wavelengths. Question: is it possible to have absorption spectra without interferences in biological samples with your method?

L328: there is no difference of 16 u between M7 and M10 nor for parent ions (26 u) nor for fragment ones, maybe the authors mean M6 after correction of MS/MS values? / absorption peaks were

L332: the difference of 16 u between M7 and M8 is only for precursor ions, not for fragment ones (but the values may be wrong for M8…)

L337: idem than previously

L342: there is a change in parity for MS/MS ions, which means a radical loss and a radical anion. So the difference of 32 u is only true for parent ions. The authors mentioned an ion at m/z 229.0167 for ellagic acid which appears for P4 and not for M3?

L348:3.19 ppm: this value is different from the one of Table 2 / In the Table, fragments of M11 have odd values (radicals) not even ones, corresponding to a loss of CH3 (15 Da) / m/z 124 (C6H4O3-) (-CO2). Even if in some spectra the authors seem to lose CH2, in practice it’s quite difficult on such molecules.

L352: literature

L355: parent ions have a difference of 14 u, not fragment ones (check values for M12 MS/MS ions).

L358: m/z 241.2170: this value is different than the one from Table 2 / C13H6O4- / CH3 (15 Da)

L364: II phase -> phase II? (many occurrences)

L366: MS/MS² -> MS/MS or MS² / fragment ions

L369: fragment ion m/z 433.0787 [M-H]- was obtained

L371: a methyl radical: here it’s not a radical loss but a neutral one

L374: secondary

L377: 299.3491 (-16 Da): the digits are abnormal / rest of the sentence to correct and there is no methyl to remove on ellagic acid.

L382: secondary fragments -> molecular ion

L385: UHPLC -> HPLC / m/z 322.9272: this value is different than the one of Table 2

L388, 403: metabolite

L392: metabolites

L399: fragment ions were / the mass difference between 306 and 227 is 79, not 80: check values

L404: C14H9O5- / 0.11 ppm: I obtain more than 28 ppm when I check the mass accuracy / C13H7O5- / C12H7O4-

L432, 433, 458: “the loss of CH2” / “Ch2 loss”: I cannot find any loss of CH2 on Figure 5 nor on Figure 6

Figure 6: there is “Plasma” in the legend, but not on the Figure

L460: Ultra-high -> High

L468: and its aglycone urolithin A

General: case changes

There is a space between a number and the unit (0.01 M, 0.22 µm, 44 Da for example) except for °C.

There is no space between °C and the number (4°C for example). Please always use the same character throughout the article.

Author Response

Brief summary:

The manuscript deals with the identification by HPLC-HRMS of metabolites from Sanguisorba tannins after rats in vivo and in vitro studies.

General comments:

The authors have to carefully check the tables which list the metabolites because there are a lot of mistakes (wrong m/z or formula). So, they are some inconstancies between their tables and their interpretation in section 3.4 of the manuscript.

Dear reviewer, thank you for your valuable comments, I am sorry for my stupid negligence, I have responded to your request.

Specific comments:

Title: UHPLC -> HPLC. The authors may have a UHPLC system (not described in the material and method) but they use an HPLC column. This should be corrected along the whole document.

Dear reviewer, thank you for your valuable comments, I am sorry for my stupid negligence, I have responded to your request.

L.41: poly merized -> polymerized

Dear reviewer, thank you for your valuable comments, I am sorry for my stupid negligence, I have responded to your request.

L.87: Welch Ultimate XB-C18… -> Welch Ultisil XB-C18 HPLC column (4.6x250 mm, 5 µm) (it could be an Ultimate column, but I could not find the reference)

Dear reviewer, thank you very much for your valuable comments. After verification, we are using this column. It may be domestically produced, which is relatively rare in the references.

L.91: 2 µL -> Two microliters

Dear reviewer, thank you for your valuable comments, I am sorry for my stupid negligence, I have responded to your request.

L93: electro spray -> electrospray

Dear reviewer, thank you for your valuable comments, I am sorry for my stupid negligence, I have responded to your request.

L.94: the authors present MS conditions for positive and negative ionization together, which gives a wrong readability. In fact, it seems they only use negative mode, so they can reduce to negative parameters.

Dear reviewer, thank you for your valuable comments, I am sorry for my stupid negligence, I have responded to your request.

L94, 95: arb. ->      A.U.

Dear reviewer, thank you for your valuable comments, I am sorry for my stupid negligence, I have responded to your request.

L96: tube lense -> tube lens

Dear reviewer, thank you for your valuable comments, I am sorry for my stupid negligence, I have responded to your request.

L98: 0.25 q of … activation time -> these parameters are not needed, except if authors do not use default parameters. However, authors should mention if they analyzed MS/MS spectra with the LTQ or with the Orbitrap (I guess it was the second solution, since there was a lot of digits in tables although they were sometimes far from the theoretical values). Authors should also mention the resolution and scan time at which they set the Orbitrap for MS and MS/MS acquisitions and if possible the parameters for Data Dependent Acquisition (because I suppose they use DDA to perform MS/MS of the metabolites).

Dear reviewer, thank you for your valuable comments, I am sorry for my stupid negligence, I have responded to your request.

L100: Please remove “and the software is freely available”: Xcalibur is a licensed software and people have to pay to use it after 2 or 3 months of free trial.

Dear reviewer, thank you for your valuable comments, I am sorry for my stupid negligence, I have responded to your request.

L116: 400macroporous -> 400 macroporous / 16 x 300 -> 16x300

Dear reviewer, thank you for your valuable comments, I am sorry for my stupid negligence, I have responded to your request.

L123: Foline-Ciocalteu -> Folin-Ciocalteu (and add hyphen in the next line)

Dear reviewer, thank you for your valuable comments, I am sorry for my stupid negligence, I have responded to your request.

L133, L260: 65.52% -> 65.5%

Dear reviewer, thank you for your valuable comments, I am sorry for my stupid negligence, I have responded to your request.

L159: in vivo

Dear reviewer, thank you for your valuable comments, I am sorry for my stupid negligence, I have responded to your request.

L162 and following: rpm/min -> rpm but ideally the speed of a centrifuge should be expressed in g (because the reader does not know the lever arm of the author’s apparatus). At L202, the speed is correctly expressed (if it’s really g). L203: there should be an error: 10,5000 g

Dear reviewer, thank you for your valuable comments, I am sorry for my stupid negligence, I have responded to your request.

L169: activated -> conditioned

Dear reviewer, thank you for your valuable comments, I am sorry for my stupid negligence, I have responded to your request.

L170: Then urine … 3 mL methanol -> Then urine was loaded on the SPE cartridge, rinsed with 5 mL of deionized water and eluted with 3 mL of methanol.

Dear reviewer, thank you for your valuable comments, I am sorry for my stupid negligence, I have responded to your request.

L184: after 4h after -> 4h after

Dear reviewer, thank you for your valuable comments, I am sorry for my stupid negligence, I have responded to your request.

L197: in vitro

Dear reviewer, thank you for your valuable comments, I am sorry for my stupid negligence, I have responded to your request.

L208, 211: units/mL -> U/mL

Dear reviewer, thank you for your valuable comments, I am sorry for my stupid negligence, I have responded to your request.

L209, 219: Tris/KCL -> Tris/KCl

Dear reviewer, thank you for your valuable comments, I am sorry for my stupid negligence, I have responded to your request.

L228: ml -> mL

Dear reviewer, thank you for your valuable comments, I am sorry for my stupid negligence, I have responded to your request.

L236, 261: was -> were

Dear reviewer, thank you for your valuable comments, I am sorry for my stupid negligence, I have responded to your request.

L263: its -> their

Dear reviewer, thank you for your valuable comments, I am sorry for my stupid negligence, I have responded to your request.

Table 1: RT[min] -> tR(min) (to harmonize with other tables); what is the difference between ‘Group areas” and “Area”? Is this information useful? There are some mistakes in the molecular weight of the molecules: if authors would express neutral molecular weight as exact mass, it should be 484.0853; 442.0900 and 634.0806. I think the last one should correspond to an [M-H]- ion. For MS2 experiments, authors should mention the polarity (at least in legend) and ideally the relative intensities of the fragments (idem for the others tables).

Dear reviewer, thank you for your valuable comments, I am sorry for my stupid negligence, I have responded to your request.

Figure 1 could be more informative if BPC was used instead of TIC and if the EIC of compounds 1, 2 and 3 were overlaid because they are not visible on the chromatograms. Since the positive mode was not any more used (a priori), it could be removed to expand the view of the negative one. If both chromatograms are kept, the legend should be enriched: lot of people are not familiar with Xcalibur and does not know where the polarity is indicated.

Dear reviewer, thank you for your valuable comments, I am sorry for my stupid negligence, I have responded to your request.

Figure 2: the upper arrows are disturbing and should be replaced by simple lines. Other lines should be added from the different boxes to the horizontal line (from yellow boxes to the blue one and from blue boxes to the yellow one)

Dear reviewer, thank you for your valuable comments, I'm very sorry that my clumsy drawing made you feel uncomfortable, I have re-edited the figure according to your opinion

L269, 273: Sanguisorba

Dear reviewer, thank you for your valuable comments, I am sorry for my stupid negligence, I have responded to your request.

L270: LTQ-Orbitrap-MS: it’s not necessary to explain LTQ since it’s the name of the first analyzer. The authors explain CID and ESI in the legend, but these abbreviations are not present on the figure.

Dear reviewer, thank you for your valuable comments, I am sorry for my stupid negligence, I have responded to your request.

Figure 3: hexopyranos -> hexopyranose

Dear reviewer, thank you for your valuable comments, I am sorry for my stupid negligence, I have responded to your request.

Tables 2, 3 and 4: the authors should mention the kind of ions ([M-H]-) either in the title bar (ideally) or in the legend. For MS2 spectra it should be nice to have relative intensities into brackets (257.0106 (100); 229.0167(80) for example). Since the metabolites have been classified by classes rather than by tR, it’s sometimes difficult to follow.

Dear reviewer, thank you for your valuable comments, I am sorry for my stupid negligence, I have responded to your request.

There are some differences between the values present in the tables and the ones in the text: the authors will have to check all values and formula for accuracy; sometimes the mass accuracy (ppm) seems to be wrong too. Note: when there is a change in parity, it means you have a radical ion. Since you have no nitrogen, all odd masses are ions and all even masses correspond to radical ions.

Dear reviewer, thank you for your valuable comments, I am sorry for my stupid negligence, I have responded to your request.

M6: m/z 241 for MS/MS ?

M8: m/z 226 for MS/MS ?

M12: MS/MS values to check

M16: C20H18O11

M17: C21H18O11

M18: C15H8O8 / m/z 299.3491: digits are too far from expected/possible values

M20: C13H8O8S / m/z 243.5576: digits are too far from expected/possible values (it looks likes a multicharged ion) / m/z 323.9272: there should be an error

N8: C13H8O7S / m/z 306.0312: there should be an error

Dear reviewer, thank you for your valuable comments, I am sorry for my stupid negligence, I have responded to your request.

There are some differences in retention times for the same metabolites. Can the authors explain if it’s due to the biological matrix or because the analysis were performed at different times? Have the authors spiked some samples with standards 1, 2 and 3 to check retention times and fragmentation spectra. Have the authors checked the presence or absence of the different metabolites in the plant?

Dear reviewer, thank you for your valuable comments, because of the biological matrix is different, there are some differences in retention times for the same metabolites. And we didn’t checked the presence or absence of the different metabolites in the plant.

Figure 4: Sanguisorba, in vivo, in vitro (x2)

Dear reviewer, thank you for your valuable comments, I am sorry for my stupid negligence, I have responded to your request.

There is a “free” OH on the left of the figure. Why are some glucuronide localized on the molecule and not mentioned as the other without a known position? For a better readability, it would be better if all molecules could be drawn in the same orientation (especially for Figure 5)

Dear reviewer, thank you for your valuable comments, I am sorry for my stupid negligence, I have responded to your request.

Figure 5: …metabolite structures in vivo and in vitro (x2)

hecked the presence or absence of the different metabolites in the plant

The authors should draw all molecules with the same orientation and the same scale. For a better readability, the should more space the structure and avoid overlay with the arrows.

Dear reviewer, thank you for your valuable comments, I am sorry for my stupid negligence, I have responded to your request.

In the upper left of the figure, it looks that it’s twice the same molecule (M3)

M3 -> M4 (e)

M17 -> M1 (b+e)???

Why some glucuronides are “GluA” and some other with developed structure, which takes more space?

L292: C7H5O5-

Dear reviewer, thank you for your valuable comments, I am sorry for my stupid negligence, I have responded to your request.

L296: successive (many occurrences along the text). It may be difficult to lose CO2 on pyrogallol in ESI since it means breaking aromaticity (easier and possible in EI) but if you observe the same thing on standard, maybe.

Dear reviewer, thank you for your valuable comments, after careful discussion with the teacher and other authors, the identification of the compound is correct. Thank you again for your valuable comments.

L300: C14H3O7- and the theoretical mass is m/z 282.9884, so the experimental one is quite far. The reference 30 (L303 and 310) does not bring information for elucidation with MS spectra

Dear reviewer, thank you for your valuable comments, I am sorry for my stupid negligence, I have responded to your request.

L304: deprotonated / Formatting of the formula / The loss of water may come from the lactone rather than from a phenol where the OH is more difficult to fragment from the aromatic ring. / Attention: 275-229 = 46: H2O then CO ≠ CO2 (44). Cf. what you obtain for M6 / indicted -> indicated? (+L456)

Dear reviewer, thank you for your valuable comments, I am sorry for my stupid negligence, I have responded to your request.

L311: Metabolites M5 and M6 / was -> were (L315) / absorption spectra -> absorption peaks? The authors say “two absorption” and described three wavelengths. Question: is it possible to have absorption spectra without interferences in biological samples with your method?

Dear reviewer, thank you for your valuable comments, after careful discussion with the teacher and other authors, the identification of the compound is correct. Thank you again for your valuable comments.

L328: there is no difference of 16 u between M7 and M10 nor for parent ions (26 u) nor for fragment ones, maybe the authors mean M6 after correction of MS/MS values? / absorption peaks were

Dear reviewer, thank you for your valuable comments, I am sorry for my stupid negligence, I have responded to your request.

L332: the difference of 16 u between M7 and M8 is only for precursor ions, not for fragment ones (but the values may be wrong for M8…)

Dear reviewer, thank you for your valuable comments, I am sorry for my stupid negligence, I have responded to your request.

L337: idem than previously

Dear reviewer, thank you for your valuable comments, I am sorry for my stupid negligence, I have responded to your request.

L342: there is a change in parity for MS/MS ions, which means a radical loss and a radical anion. So the difference of 32 u is only true for parent ions. The authors mentioned an ion at m/z 229.0167 for ellagic acid which appears for P4 and not for M3?

Dear reviewer, thank you for your valuable comments, I am sorry for my stupid negligence, I have responded to your request.

L348:3.19 ppm: this value is different from the one of Table 2 / In the Table, fragments of M11 have odd values (radicals) not even ones, corresponding to a loss of ●CH3 (15 Da) / m/z 124 (C6H4O3●-) (-CO2). Even if in some spectra the authors seem to lose CH2, in practice it’s quite difficult on such molecules.

Dear reviewer, thank you for your valuable comments, I am sorry for my stupid negligence, I have responded to your request.

L352: literature

Dear reviewer, thank you for your valuable comments, I am sorry for my stupid negligence, I have responded to your request.

L355: parent ions have a difference of 14 u, not fragment ones (check values for M12 MS/MS ions).

Dear reviewer, thank you for your valuable comments, I am sorry for my stupid negligence, I have responded to your request.

L358: m/z 241.2170: this value is different than the one from Table 2 / C13H6O4- / ●CH3 (15 Da)

Dear reviewer, thank you for your valuable comments, I am sorry for my stupid negligence, I have responded to your request.

L364: II phase -> phase II? (many occurrences)

Dear reviewer, thank you for your valuable comments, I am sorry for my stupid negligence, I have responded to your request.

L366: MS/MS² -> MS/MS or MS² / fragment ions

Dear reviewer, thank you for your valuable comments, I am sorry for my stupid negligence, I have responded to your request.

L369: fragment ion m/z 433.0787 [M-H]- was obtained

Dear reviewer, thank you for your valuable comments, I am sorry for my stupid negligence, I have responded to your request.

L371: a methyl radical: here it’s not a radical loss but a neutral one

Dear reviewer, thank you for your valuable comments, I am sorry for my stupid negligence, I have responded to your request.

L374: secondary

Dear reviewer, thank you for your valuable comments, I am sorry for my stupid negligence, I have responded to your request.

L377: 299.3491 (-16 Da): the digits are abnormal / rest of the sentence to correct and there is no methyl to remove on ellagic acid.

Dear reviewer, thank you for your valuable comments, I am sorry for my stupid negligence, I have responded to your request.

L382: secondary fragments -> molecular ion

Dear reviewer, thank you for your valuable comments, I am sorry for my stupid negligence, I have responded to your request.

L385: UHPLC -> HPLC / m/z 322.9272: this value is different than the one of Table 2

Dear reviewer, thank you for your valuable comments, I am sorry for my stupid negligence, I have responded to your request.

L388, 403: metabolite

Dear reviewer, thank you for your valuable comments, I am sorry for my stupid negligence, I have responded to your request.

L392: metabolites

Dear reviewer, thank you for your valuable comments, I am sorry for my stupid negligence, I have responded to your request.

L399: fragment ions were / the mass difference between 306 and 227 is 79, not 80: check values

Dear reviewer, thank you for your valuable comments, I am sorry for my stupid negligence, I have responded to your request.

L404: C14H9O5- / 0.11 ppm: I obtain more than 28 ppm when I check the mass accuracy / C13H7O5- / C12H7O4-

Dear reviewer, thank you for your valuable comments, I am sorry for my stupid negligence, I have responded to your request.

L432, 433, 458: “the loss of CH2” / “Ch2 loss”: I cannot find any loss of CH2 on Figure 5 nor on Figure 6

Dear reviewer, thank you for your valuable comments, I am sorry for my stupid negligence, I have responded to your request.

Figure 6: there is “Plasma” in the legend, but not on the Figure

Dear reviewer, thank you for your valuable comments, I am sorry for my stupid negligence, I have responded to your request.

L460: Ultra-high -> High

Dear reviewer, thank you for your valuable comments, I am sorry for my stupid negligence, I have responded to your request.

L468: and its aglycone urolithin A

Dear reviewer, thank you for your valuable comments, I am sorry for my stupid negligence, I have responded to your request.

General: case changes

There is a space between a number and the unit (0.01 M, 0.22 µm, 44 Da for example) except for °C.

Dear reviewer, thank you for your valuable comments, I am sorry for my stupid negligence, I have responded to your request.

There is no space between °C and the number (4°C for example). Please always use the same character throughout the article.

Dear reviewer, thank you for your valuable comments, I am sorry for my stupid negligence, I have responded to your request.

Round 2

Reviewer 1 Report

The authors did not revise properly my previous comments:

-Results and discussion could be separately written for better understanding

-All figures to be regenerated using Arial fonts with high resolution (at least figure 2 and 6)

-More limitations could be added

Author Response

Dear reviewer,

Thanks for your precious comments for the manuscript. According to your advice, we amended the relevant part in manuscript.

Reviewer 3 Report

Authors have revised the manuscript, however majority of my  recommendations were not addressed correctly:

  1. The reference 3 provided by authors is not support the phrase "...The dried root of Sanguisorba is recorded in various versions of ...European Pharmacopoeia, Russian Pharmacopoeia...". Yu et al. (2011) have not mentioned the use of Sanguisorba officinalis in European and Russian Pharmacopoeia. The most suitable references were recommended in previous round (see https://doi.org/10.15835/nsb649471; https://doi.org/10.1016/j.jep.2020.113685)
  2. In Sect. 3.1.: Please provide information who have identified a plant material and indicate the number of voucher of specimens.
  3. In Sect.3.5:  please justify a dose of tannins for peroral administration. TAuthors provide information that "The rats in vivo experiments group were orally administered to the rats at a single dose of 150 mg/kg (crude drug weight/rat weight)." and "The rats in vitro experiments group, liver microsomes and cytosol, intestinal microbiota were incubated with Sanguisorba tannins (100 mg/mL, suspended in saline).". However, it is not clear, why these doses were selected and why doses for in vivo and in vitro were different?
  4. In Section 2.1: the phrase  :'...metabolite templates (known and identified metabolites) were summarized and established using the reported metabolic transforms of ingredients in literature..." require support by appropriate references. In previous round I have suggested next literature sources about identification  of similar tannins (see https://doi.org/10.1021/acs.jafc.8b02115;         https://doi.org/10.1080/14786419.2014.923999). Actually in both  papers dry raw plant materials were used. Please compare your results and methods with others.
  5. In Fig.1 please indicate the peaks relevant to compounds from Table 1
  6. Tables 2-4: please distinguish Phase I and Phase II metabolites. I don't see any modifications.
  7. Tannins are in focus of many studies. The discussion is weak. More literature data should be discussed. (see recommended articles in question 4 and many others).
  8. Please don't use common phrase "I have modified the original text according to your requirements" in responses, but address the question indicating lines in revised versuon or provide full text response.

Author Response

(The authors gave the same response as above.)

Reviewer 4 Report

The revised one seems ok.

Author Response

Dear reviewer,

Thanks for your precious comments for the manuscript.  We will try our best to revise my manuscript.

Reviewer 5 Report

General comments:

The authors have made a lot of corrections but they still have to carefully check the accuracy of all masses in the Tables and the coherence with the text.

Specific comments:

L13: traporbitrap -> trap orbitrap

Table 1: the mass for corilagin is still wrong

Table 2: there are still some errors in masses for MS2 spectra. For example, m/z 226 and 198 for M8, but there may be other ones (I have not check all masses). Authors have to carefully check all measured masses of the Tables

Table 3: the mass for N8 is still wrong

Figure 5: M3 -> M4: decarbonylation rather than decarboxylation (-CO). I still have a problem with the arrow from M17 to M1. M7 -> M15: it’s the same molecules on the Figure and the arrow mentioned “methylation” but on Figure 4, M15 is a glucuronic compound. The representation of M16 is different between Figures 4 and 5 (the molecular formula is the same but the position of the modifications is different)

Section 2.4: a lot of corrections have been made but some errors remains:
“successive elimination”: you need to have many fragmentations to have successive ones (at least two)
fragments ion -> fragment ions. Sometimes authors use “fragment ions” but I think they mean “parent ions” L153 and L157 for example
L133: 26 Da -> 28 Da

L211: lost -> loss

Section 3.3: the authors have removed “positive mode” from the instrument parameters, since they don’t use it for this article, but they keep a mixture of positive and negative parameters in the description of the mass spectrometer.

Author Response

(The authors gave the same response as above.)
